# Client Applications and Server-Side Docker for Management of RNASeq and/or VariantSeq Workflows and Pipelines of the GPRO Suite

**DOI:** 10.3390/genes14020267

**Published:** 2023-01-19

**Authors:** Ahmed Ibrahem Hafez, Beatriz Soriano, Aya Allah Elsayed, Ricardo Futami, Raquel Ceprian, Ricardo Ramos-Ruiz, Genis Martinez, Francisco Jose Roig, Miguel Angel Torres-Font, Fernando Naya-Catala, Josep Alvar Calduch-Giner, Lucia Trilla-Fuertes, Angelo Gamez-Pozo, Vicente Arnau, Jose Maria Sempere-Luna, Jaume Perez-Sanchez, Toni Gabaldon, Carlos Llorens

**Affiliations:** 1Biotechvana, Parc Científic Universitat de València, 46980 Paterna, Spain; 2Faculty of Biomedical and Life Sciencies, Universitat Pompeu Fabra, 08002 Barcelona, Spain; 3Faculty of Computers and Information, Minia University, Minia 61519, Egypt; 4Institute of Aquaculture Torre de la Sal (IATS), Consejo Superior de Investigaciones Cientificas (CSIC), 12595 Castellón, Spain; 5Institute for Integrative Systems Biology (I2SysBio), Universitat de Valencia y CSIC, UVEG-CSIC, 46980 Paterna, Spain; 6Valencian Research Institute for Artificial Intelligence (VRAIN-UPV), Universitat Politècnica de València, 46022 Valencia, Spain; 7Genomics Unit Cantoblanco, Parque Científico de Madrid, 28049 Madrid, Spain; 8Facultad de Ciencias de la Salud, Universidad San Jorge, 50830 Zaragoza, Spain; 9Escola Tècnica Superior d’Enginyeria, Universitat de València, 46100 Burjassot, Spain; 10Biomedica Molecular Medicine SL, Parque Científico de Madrid, 28049 Madrid, Spain; 11Molecular Oncology and Pathology Lab, Hospital Universitario La Paz-IdiPAZ, 28046 Madrid, Spain; 12Institució Catalana de Recerca i Estudis Avançats, 08010 Barcelona, Spain; 13Life Sciences, Barcelona Supercomputing Centre (BSC-CNS), 08034 Barcelona, Spain; 14Mechanisms of Disease Programme, Institute for Research in Biomedicine (IRB), The Barcelona Institute of Science and Technology, 08028 Barcelona, Spain

**Keywords:** RNASeq, VariantSeq, server-side, pipelines, workflows, resequencing, interface environments, artificial intelligence

## Abstract

The GPRO suite is an in-progress bioinformatic project for -omics data analysis. As part of the continued growth of this project, we introduce a client- and server-side solution for comparative transcriptomics and analysis of variants. The client-side consists of two Java applications called “*RNASeq*” and “*VariantSeq*” to manage pipelines and workflows based on the most common command line interface tools for RNA-seq and Variant-seq analysis, respectively. As such, “*RNASeq*” *and* “*VariantSeq*” are coupled with a Linux server infrastructure (named GPRO Server-Side) that hosts all dependencies of each application (scripts, databases, and command line interface software). Implementation of the Server-Side requires a Linux operating system, PHP, SQL, Python, bash scripting, and third-party software. The GPRO Server-Side can be installed, via a Docker container, in the user’s PC under any operating system or on remote servers, as a cloud solution. “*RNASeq*” and “*VariantSeq*” are both available as desktop (RCP compilation) and web (RAP compilation) applications. Each application has two execution modes: a step-by-step mode enables each step of the workflow to be executed independently, and a pipeline mode allows all steps to be run sequentially. “*RNASeq*” and “*VariantSeq*” also feature an experimental, online support system called GENIE that consists of a virtual (chatbot) assistant and a pipeline jobs panel coupled with an expert system. The chatbot can troubleshoot issues with the usage of each tool, the pipeline jobs panel provides information about the status of each computational job executed in the GPRO Server-Side, while the expert system provides the user with a potential recommendation to identify or fix failed analyses. Our solution is a ready-to-use topic specific platform that combines the user-friendliness, robustness, and security of desktop software, with the efficiency of cloud/web applications to manage pipelines and workflows based on command line interface software.

## 1. Introduction

Advances in next-generation sequencing (NGS) have changed the way researchers perform comparative analyses based on RNA-seq and variant-seq resequencing data (for an example review, see [1]). Implementing these approaches into routine laboratory procedures remains challenging, as they require the sequential execution of complex and variable protocols to extract and process the biologically relevant information from the raw sequence data. These protocols are typically called pipelines and/or workflows and are usually carried out using command line interface (CLI) software. The advantage of pipelines based on CLI tools is that they can be customized for specific goals and can utilize the wide range of freely available software produced by the scientific community. This is particularly useful in RNA-seq and variant-seq approaches, where the requirements of each pipeline differ depending on the data to be analyzed [2,3,4,5,6]. For example, RNA-seq pipelines vary depending on the availability of GTF/GFF files (the file format that provides information about the gene features of a reference sequence) and the reference sequence (it can be a genome, a transcriptome, a gene panel, etc.). Similarly, variant-seq pipelines vary depending on the type of variants (single-point mutations, indels, etc.) or according to the source and frequency of the target variants (somatic or germinal). Another advantage of protocols based on CLI tools is that they run on both personal computers (PCs) and remote servers. This allows the simultaneous management and analysis of multiple samples, a practice that is typical in RNA-seq and variant-seq approaches. The disadvantages of pipelines based on CLI tools is, however, that their implementation and usage can only be achieved on Linux environments and requires advanced informatic skills for installing third-party software, writing scripts, and executing processes with the command line. In other words, these protocols are restricted to experienced bioinformaticians.

In recent years, distinct Graphical User Interface (GUI) solutions have been developed to provide end-users with friendly-to-use tools to autonomously analyze their NGS data. Many of these are commercial platforms distributed under payment licenses, while others are free or publicly available tools. Commercial platforms featuring functions for comparative transcriptomics and variant analysis are usually desktop cross-platforms that run in any PC and operative system (OS) but can also be web platform providers of cloud services. While some users consider these commercial platforms overpriced, others find them worth their value because: (i) they present specific proprietary tools; (ii) their management only requires informatic skills at the user level; and (iii), they are implemented under very efficient and secure Java or C++ frameworks. Examples of popular commercial solutions with tools for the analysis of RNA-seq and variant-seq data are Qiagen (CLC) OmicSoft [7], Geneious [8], Partek [9], and OmicsBox [10]. In spite of their benefits, commercial packages remain limited in comparison with pipelines based on CLI tools that can be more easily updated and upgraded with new CLI tools than commercial platforms. For this reason, there is a growing tendency among bioinformatic companies to enable plugins for third-party CLI software in their platforms. On the other hand, most of the free or publicly available GUI solutions for comparative transcriptomics and variant analysis are platforms powered by CLI software. These platforms are usually implemented as client- and server-side solutions where the client-side component is the desktop and/or web application allowing the CLI tools to be run via GUI, and the server-side component is the platform hosting the CLI tools and that provides the Linux and R environments allowing the CLI tools to run. Two representative examples of already assembled client- and server-side public solutions are Unipro UGENE [11] and Chipster [12]. UGENE is a cross-platform developed in C++ with distinct tools for molecular biology analysis, including pipelines and components based on the most common CLI tools for distinct -omics [13]. Chipster is another platform providing GUI environments for a collection of CLI tools for the analysis of data from distinct -omics (RNA-seq and variant-seq included). Chipster was originally developed as a desktop Java application but is currently available as a web server. Chipster can also be installed on the user’s PC, albeit only under Linux environments, because Chipster is a client application coupled to a bundle containing all server-side dependencies; mainly, because CLI software only runs on Linux OS. UniGene and Chipster are open source solutions permitting their users to implement new tools and some levels of personalization to configure specific workflows. These platforms are indeed a valuable resource for users with advanced bioinformatic backgrounds, but they could be complex to deal with, and therefore challenging, for other users whose expertise and background is more biological. Client- and server-side GUI solutions based on CLI tools can also be implemented de novo. These personalized approaches are usually implemented as web instances using Python or R because they are interpreted languages that are more concise and easier to manage than Java and C++. In the case of R developers, the most common strategy for this kind of implementation is to combine RStudio [14], an integrated environment for R programming, with Shiny [15], a framework to implement web server solutions based on R tools. As for Python, developers usually use frameworks such as Bioconda [16] and Galaxy [17], a platform of precompiled web modules acting as GUIs for specific CLI tools. Galaxy modules can be combined in different ways to construct personalized workflows for RNA-seq and variant-seq, or for any other -omics topic (the repertoire of tools supported by the Galaxy project is very extensive). However, implementing a Galaxy solution remains convoluted, as installing and configuring specific combinations of Galaxy modules is a complex job that requires advanced bioinformatic skills with a significant background in informatic systems. Moreover, Galaxy only runs on Linux environments, mainly because the CLI tools used as server-side dependencies are Linux tools. However, advanced users interested in running a Galaxy approach under other OS, such as Windows or MacOS, have the option to create Docker containers [18] to deploy therein the Galaxy server-side component. Additionally, Galaxy is also available as a public web service at (https://usegalaxy.org, accessed on 17 January 2023).

In this article, we are pleased to introduce a new client- and server-side solution allowing end-users to perform comparative transcriptomics and variant analysis, autonomously, using CLI tools under friendly GUI environments. Our solution is based on two client Java applications named “*RNASeq*” and “*VariantSeq*” that are coupled to a server platform called GPRO Server-Side (GSS). The GSS contains all dependencies needed by each application to run their pipelines and workflows. This solution belongs to the GPRO suite, a bioinformatic project for -omics data analysis whose initial release [19,20] was a multi-task desktop application with client functions to perform functional analyses via cloud computing strategies. The second and current GPRO release consists of a suite of six applications (“*SeqEditor*”, “*RNASeq*”, “*VariantSeq*”, “*DeNovoSeq*”, “*Worksheet*”, and “*STATools*”), each devoted to a specific topic. Briefly, “*SeqEditor*” is an application for molecular sequence analysis that was recently introduced in a previous article [21]. As explained above, “*RNASeq*” and “*VariantSeq*” are two independent client applications published in this article and specifically oriented for RNA-seq and variant-seq analysis, the two topics that give the applications their names). The three other applications (“*DeNovoSeq*”, “*Worksheet*”, and “*STATools*”) are devoted to other topics and will be published in forthcoming articles (for more information about the GPRO suite, see https://gpro.biotechvana.com, accessed on 17 January 2023). The GPRO suite also features a smart experimental artificial intelligence system for user support called GENIE, which is also introduced in this article.

## 2. Material and Methods

### 2.1. Client-Side Applications

The framework of “*RNASeq*” and “*VariantSeq*” was developed in Java 11 and Desktop, and Cloud versions were created using Eclipse Rich Client Platform 4.22 (RCP) and Eclipse Remote Application Platform 3.22 (RAP), respectively [22]. The implementation of this framework follows a similar approach to the model–view–controller (MVC) pattern [23]. At the model layer, the framework includes all implementations needed to represent the low-level elements of the tools’ wrapper descriptor (e.g., JobDescriptor and different types of VariableDescriptor such as input files and tools parameters), as well as workflow templates’ descriptors. At the view layer, we implemented automated utilities to generate GUIs for single tasks or workflows within each CLI tool using the selected JobDescriptor or WorkflowTemplates. At the controller level, the implementation includes task or workflow instances controlling and storing user inputs captured by the GUIs based on the model layer, and that are also responsible for executing and tracking the tasks on the GSS. As part of the workflow framework at the controller layer, the Bash framework validates the tasks from the user side and generates bash scripts from tasks’ descriptors, submitting them to the GSS for running. In such scripts, tracking events are inserted to track general tasks, check the status of running tasks, and collect log files. All events are stored in user space on the GSS and sent back to the client’s applications for visualization.

### 2.2. GPRO Server-Side Platform

GSS is a Linux infrastructure that hosts all the dependencies required by “*RNASeq*” and “*VariantSeq*” to run pipelines and workflows on the Server-Side. GSS constitutes the following elements:Linux Operating System with at least Bash version 4;MySQL Server for installing databases;Apache HTTP server 2.2 or later;PHP 7 or later;R 3.6.0 or later and a compatible Bioconductor with R version;Perl 5;Python 2.7;Third-party CLI software (see Table 1 for details);An API for communicating between client applications and GSS.

The installation of GSS requires complex steps to setup Linux, Apache, MySQL, and PHP (LAMP stack), as well as the CLI software. It also requires scripts for handling the incoming requests to GSS that must be manually installed. To overcome this, we deployed GSS in a Docker container [18] that can be easily installed on remote servers or any PC or Mac using the OS, Windows, or Linux, as long as there is sufficient disk space and RAM. The minimum requirements are 500 Gb of hard disk and 16Gb of RAM.

### 2.3. Virtual Chatbot Assistant and Expert System

“*RNASeq*” and “*VariantSeq*” are supported by an experimental artificial intelligence (AI) system called GENIE, which was created and trained using natural language processing and machine learning methodologies [49,50]. GENIE consists of distinct interfaces, dialogs, and scripts (the client-side part) that are linked to a server-side module composed of the following elements: (1) knowledge databases; (2) the expert system; and (3) the virtual (chatbot) assistant. These three features are centralized in a GPRO remote server so that the expert system and the chatbot can be continually fed new training data. Below is a detailed description of each element.

Knowledge databases: The chatbot and the expert system are supported by five knowledge databases that are shared between the virtual assistant and the expert system:Questions and answers database. This database identifies and stores key terms and serves as an index of answers to different questions.CLI tools dependency database. This database stores information on the type of input that each CLI tool receives and the output that it generates, as well as information on different parameters and customization options.Contextual database. This database provides a graphical representation to all pipelines/workflows and the programs implemented in each protocol.Key terms database. This is a database of generic questions about different protocols or programs.Log files database. This is a database that stores the information reported by the log files generated by the CLI software dependencies.

Information was taken from the “*RNASeq*” and “*VariantSeq*” manuals (available at the Section “Data Availability Statement”) and from public scientific networks and/or repositories such as Biostar [51], SeqAnswers [52], Pubmed [53], and the GATK community forum (https://gatk.broadinstitute.org/hc/en-us/community/topics, accessed on 6 June 2022).

Expert system: This is a rule-based system that provides users with actionable solutions for troubleshooting problems in failed analyses. The expert system was implemented in Python using the Django framework 3.2.8 (https://www.djangoproject.com, accessed on 17 January 2023) and trained using machine learning methodologies [54,55]. It consists of:Inference engine: This handles the users’ request by processing the logs and the tracking information sent by the job tracking panel of client applications, with the objective of extracting key features and errors information that can be used to query the solutions database.Proven facts database: This database contains the rules managed by the inference engine for recommendations of how to fix problems and errors from failed analyses.Administration panel: This is a website provided for administration and management of the expert system when applying rules or adjusting aspects, such as adding new task descriptors, editing databases, managing actions/recommendation templates, etc. The administration panel is only accessible by experts from our side or by users interested in contributing to the training of this tool.Client interface: This is the interface implemented in the pipeline jobs panel of the client applications (“*RNASeq*” and “*VariantSeq*”) to manage the interaction with the expert system engine.API: The API allows the interface to accept requests from the client applications and enables client applications to track and fetch the actions/recommendations proposed by the expert system.

Chabot Engine: The chatbot helps users to resolve issues with installation, technical errors, user guides, or FAQs. The chatbot engine was implemented using python via the Rasa open-source framework 2.8.16 [49] and pre-trained Universal Sentence Encoder language models [50]. The chatbot engine utilizes a Retrieval-based strategy with intent classification, entity identification and extraction, and response selection from a set of predefined responses. The chatbot is considered a level 3 conversational AI, as it can understand questions from the context and handle unexpected queries (users changing their mind, etc.). The training dataset was mainly compiled from our collection of Q/A databases focusing on client applications and bioinformatic related concepts and extended to other Q/A data sources (the above-referred knowledge databases). Users are allowed to interact with the chatbot via two different interfaces:Online Web interface available at https://gpro.biotechvana.com/genie (accessed on 17 January 2023). This webpage includes a dialog where users can ask questions and the chatbot will respond using a graphical summarization of the different protocols of each GPRO application including “*RNASeq*” and “*VariantSeq*”.An interactive user interface implemented in each client application to query the chatbot directly from the application.

The chatbot allows an API developed using the Rasa framework to modulate the communication between client applications and the chatbot.

## 3. Results

### 3.1. General Overview

“*RNASeq*” and “*VariantSeq*” are two cross-platform client applications built for the processing and analysis of resequencing data obtained via NGS technologies. Specifically, “*RNASeq*” offers a GUI-based environment to manage pipelines and workflows based on CLI tools for differential expression (DE) and enrichment analysis. “*VariantSeq*” offers a similar solution but for the calling and annotation of single-point mutations (SNP) and indels. “*RNASeq*” and “*VariantSeq*” can be installed on the user’s PC (desktop version) or used via a web browser (cloud or web version). Analyses performed by “*RNASeq*” and “*VariantSeq*” are executed in GSS via a Linux server infrastructure hosting a collection of CLI tools (Table 1) that are used by both applications as pipelines and workflow dependencies. To this extent, GSS includes an API and other server-side dependencies needed to link each client application (“*RNASeq*” or “*VariantSeq*”) to GSS. Figure 1 shows a technical schematic for the framework of “*RNASeq*” or “*VariantSeq*” and how it operates for executing single analyses or pipeline complex analyses in GSS. As the latter is a complex infrastructure, it was deployed in a docker container that can be easily installed on remote servers or the user’s PC. The current version of the GSS docker supports one or two users working simultaneously; however, we are committed to releasing a future version for servers with multiple users. Currently, servers with requirements for multiple users will have to manually install GSS tool-by-tool (server administrators interested in that possibility can contact us for more detailed information).

### 3.2. User Interface

“*RNASeq*” and “*VariantSeq*” use a common user interface (shown in Figure 2) to access the GSS and manage analyses. The user interface is structured into the following modules:“*FTP Browser*”. This is a File Transfer Protocol (FTP) to provide users access to the GSS and to transfer files/folders from the user PC to the GSS, or vice versa.“*Working space*”. This is the framework space from which the GUIs manage the CLI tools hosted at the GSS.“*Top Menu*”. This is the main menu for each application and is located at the top of the interface. All tools and tasks are organized into different tabs as detailed below:“*Directory*”. This tab is for users to select and set the main directory for exchanging material with the GSS using the FTP browser.“*Transcripts/Variant Protocols*”. This tab provides access to the modes of computation and protocols of each application. By clicking on this tab, the user can choose between two computational modes: step-by-step or pipeline. When selecting the step-by-step mode, a “*Task Menu*” appears in the working space to provide access to the set of GUIs for the distinct CLI tools and/or commands implemented in the step-by-step workflow for each application. When choosing the pipeline mode, the user accesses the pipeline manager of each application.“*Pipeline Jobs*”. This tab allows the user to track the status of all jobs executed in the GSS or to obtain recommendations from the GENIE’s expert system to troubleshoot computational issues in failed analyses.“*Preferences*”. This tab allows the user to configure and activate the connection settings between the client application and the GSS.“*Help*”. This tab provides access to the user manual for each application and to the summary panel of GENIE’s chatbot.

As previously noted, analyses are run on the Server-Side and so the client applications and GSS must be linked. To do this, users must access “*Pipeline connection settings*” in the *“Preferences”* tab and configure the connection settings as illustrated in Appendix A.

### 3.3. Protocols

“*RNASeq*” *and* “*VariantSeq*” were created based on two “good-practice” protocols for the most common and popular CLI tools in their respective topics (for more specific details, see [2,3,4,5,6]). In Figure 3, we show the protocol for DE and enrichment analysis based on which “*RNASeq*” was implemented. This protocol is based on the following steps: “*Quality Analysis & Preprocessing*”, where distinct tools for the quality analysis and preprocessing of fastq samples are provided; “*Mapping*”, offering tools to map the reads of fastq files against reference sequences; “*Transcriptome Assembly* and/or *Quantification*” to assemble and quantify the transcriptome expression patterns of the case study samples by processing the bam files obtained at the mapping step; “*Differential Expression*”, for comparison of the distinct groups/conditions under comparison; and “*Differential Enrichment*”, for assessing the differential enrichment of Gene Ontology (GO) categories and/or metabolic pathways. Two possible paths are allowed within this protocol. One path follows the “*Tophat-Cufflink*” good-practices inspired or based on the Tuxedo protocol [41], where splicing mappers such as Tophat [29] or Hisat2 [30] are combined with the Cufflinks package [41,44] to perform splicing mapping and DE analyses. These are mainly oriented (but no limited) to RNA-seq studies using genome sequence references, which are usually accompanied by GTF/GFF files. The other path is a “*Mapping & Counting*” protocol, where DNA/RNA mappers such as Bowtie [31], BWA [32] or STAR [33] are combined with tools for transcriptome quantification such as Corset [34] or HtSeq [35] to perform DE analysis with DESeq2 [42] and EdgeR [43]. This path is usually used in RNA-seq studies based on sequence references with no availability of GTF/GFF files, such as transcriptomes assembled de novo, amplicons, and gene sets. Using both paths, we consider a final enrichment analysis of GO categories and/or metabolic pathways using GOSeq [45].

“*VariantSeq*” was developed following a protocol based on the most common practices for the calling/annotation of SNP and indels using the GATK [37,38,46] and VarScan2 [47] callers and other CLI tools, including Picard [39], SAMtools [40], and others. As shown in Figure 4, the protocol of “VariantSeq” presents the following steps; “*Quality Analysis & Preprocessing*”, for fastq files preprocessing; “*Mapping*”, for mapping fastq files against reference sequences; “*Training Sets*”, for generating computational resources such as panels of normals (PON), training sets, truth sets, and known-site sets usually required to eliminate or reduce false positives; “*Postprocessing*”, for preparing bam files for the calling step (by marking duplicates, re-aligning reads/variants, adding tags, indexing data, etc.); *Variant Calling*”, for performing the variant calling using different command options of the GATK and VarScan2 packages that will vary depending on the data under analysis (genome, exome, transcriptome) and the type of variant (germinal, somatic, cancer, trio, etc.); “*Variant Filtering*”, for postprocessing VCF files generated in the variant calling steps to filter the variants according to different criteria, such as coverage, quality, frequency; and “*Annotation*”, for processing VCF files to add functional effect annotations to the called variants using the Variant Effect Predictor (VEP) of Ensembl [48].

### 3.4. Usage and Tutorials

“*RNASeq*” and “*VariantSeq*” can be executed using two different modes: step-by-step or pipeline. In step-by-step mode, analyses are executed in a stepwise manner and each analysis can be executed and/or re-executed independently from all other analyses. The “*Task Menu*” will appear in the working space with several tabs organized according to the steps of the workflow. Each tab has a scroll down sub-menu with distinct options and CLI tools to perform any analyses associated with that step. In step-by-step mode, each CLI tool has a GUI for users to declare input and output files, configure options and parameters, and run the analyses as could be achieved using the command line. Two examples of GUIs per CLI tool are provided in Appendix A (one from “*RNASeq*” and another from “*VariantSeq*”). In the pipeline mode, users can access the pipeline manager of each application to configure and run specific, sequential combinations of CLI tools. When the user accesses the pipeline manager, a summary with all possible pipeline combinations appears, allowing users to select one of these pipelines. Next, the user accesses another interface to upload the input data files and output folders, as well as to set the experiment design (identifying the groups/conditions to be compared or declaring which fastq files are replicates of a group or condition, etc). After this, users can access a pipeline menu where they can configure the options and parameters of each CLI tool associated to each step of the pipeline. Once the pipeline is configured, the user can run all the steps of the analyses in one click. In Appendix A, we present two dynamic gifs that illustrate the procedure to configure and run the respective pipeline managers of “*RNASeq*” and “*VariantSeq*”. In addition, two tutorials are available on the installation and usage of “*RNASeq*” and “*VariantSeq*” using real data from previously published works. One is a “*RNASeq*” tutorial based on a control vs. infection case study of comparative transcriptomics performed by Pérez-Sanchez et al. [56] on sea bream *Sparus aurata*. The other tutorial is for “*VariantSeq*”, which is based on a case study of cancer variant analysis previously performed by Trilla-Fuertes et al. [57] using whole-exome data sequenced from human anal squamous cell carcinoma. These two tutorials are freely available on the websites of the manuals of “*RNASeq*” and “*VariantSeq*”. A direct link to each tutorial is also provided in the section below, “*Data Availability Statement*”.

### 3.5. Smart Support System

“*RNASeq*”, “*VariantSeq*”, and GSS are linked to a smart system called GENIE that provides each application with two support tools: (i) a pipeline jobs panel powered by an expert system to monitor the status of all pipeline jobs submitted to the GSS and for providing users with recommendations to fix failed analyses; (ii) a virtual chatbot assistant to answer questions related to the usage of each application, protocols, and features of each CLI tool. In Figure 5, we provide a technical schematic of the GENIE system and screenshots of the chatbot and the pipeline jobs panel. The knowledge databases and engine cores of the chatbot and the expert system are hosted on a remote server of the GPRO project. This allows for the centralized training, growth, curation, and continual improvement of these AI systems. Each application implements dialogs and panels that interact with GENIE via API. The interface dialog for interacting with the chatbot is accessible in the “*Help*” section of each application, albeit a web version of this dialog is also available online at https://gpro.biotechvana.com/genie (accessed on 17 January 2023). The pipeline jobs panel is a dynamic register that allows the user to monitor and review the history of each job submitted to the GSS. As shown in Figure 5, this panel is structured into three screens: (i) a top screen showing all job/pipeline records submitted to the GSS; (ii) a middle screen showing all track information for a selected job record; and (iii) a bottom screen showing the log file (stdout and stderr) of the executed job. The history shown in the pipeline jobs panel is periodically updated, and users can also update this manually via the context menu. By right clicking on any history record, users have access to a contextual menu allowing the following tasks:

“*Select in FTP Explorer*”. This opens/views the output folder of the selected record.“*View Report*”. This visualizes the log file of the selected record.“*Refresh*”. This manually refreshes the history records.“*Delete*”. This deletes the selected record from the history (this only deletes the record and cached log and track information. The original files with the results are kept on the server and can only be deleted directly from the server or from the FTP Browser).“*Restart*”. This runs the analysis again with the same input data options and parameters used in the previous analysis.“*Edit & Restart*”. This runs the analysis again but allows the user to edit or modify any input data, option, or parameter from the previously used CLI tool.“*Resolve*”. This accesses the interface of the expert system, allowing the provision of recommendations on controlled actions as defined by the expert system.

With the contextual menu, the user can manage options regarding a specific job. For example, in case of a failed job, the user can re-run the analyses using the option “*Edit & Restart*”, editing first the settings and then the parameters of the analysis. If the issue persists, the user can access the expert system and try to search for a recommendation (if available) about how to solve the issue.

## 4. Discussion

This article introduces a new client- and server-side solution, developed in the context of the GPRO project, to perform comparative transcriptomics and variant analysis, using CLI tools via GUI environments. The client-side part of this solution consists of two applications named “*RNASeq*” and “*VariantSeq*”, both with cloud and desktop executables. Each application provides a customized protocol with the availability of distinct pipelines and workflows according to the topic addressed by each application, as well as two modes of execution (step-by-step and pipeline-like), and an interactive AI system called GENIE for troubleshooting. The server-side component is called GSS, a bioinformatic server infrastructure that hosts the CLI tools and all other dependencies needed by the client applications to run the analyses. The GSS is distributed as a docker container image and can be installed on a remote server or in the user’s PC. “*RNASeq*” has been successfully tested in several transcriptomics studies using distinct reference sequences and experimental backgrounds [56,58,59,60]. “*VariantSeq*” has also been validated across distinct studies for variant analysis and different experimental contexts [57,61,62]. In addition, two tutorials (one for each application) have also been created, and are here included, to provide users with training material to familiarize themselves with each application (links for the tutorials are provided in the section below, Data Availability Statement).

Comparing “*RNASeq*”, “*VariantSeq*”, and the GSS (from that point on, here referred as our solution) to other similar platforms is not straightforward, as the different solutions vary considerably in terms of functionality and features. Nevertheless, we attempt to provide the reader with an appropriate discussion about the benefits and specific niches of our solution relative to other comparable platforms that also present protocols for RNA-seq and variant-seq analysis, such as Galaxy [17], Chipster [12], and OmicSoft [7]. The first aspect to highlight about our solution is that it is a publicly available resource, as are Galaxy, UGENE, and Chipster, while OmicSoft (and similar tools) are distributed under payment license. Second, as for context and operativity, our solution is like Galaxy and Chipster as the three platforms are client- and server-side solutions aiming to provide GUI environments to run CLI software on the Server-Side. In contrast to this, OmicSoft is representative of proprietary implementations. Choosing between one type of platform or another depends on user experience. Certainly, the advances on the state-of-the-art are usually defined by public CLI software, but it is important to stress that the implementation of pipelines and/or workflows based on CLI software is currently a widespread practice among bioinformaticians, from the perspectives of both academia and industry. Moreover, bioinformatic companies are always expeditious for adapting their services and products as quickly as possible to any new trend or advance identified in the field (either developing new proprietary tools or integrating third-party software). Note, for instance, the catalog of tools and services provided by OmicSoft that includes a wide number of customized solutions for distinct omics. These are all under different implementations (desktop, cloud, workbench, etc.). Third, as for implementation, our solution is similar to OmicSoft and Chipster. The three platforms are already assembled ready-to-use resources that can be installed and used with only a couple of clicks, while Galaxy is a modular platform allowing the users to combine Galaxy modules in different ways to reach tailor-made, specific GUI solutions. Deciding between an already-assembled platform or a modular approach like Galaxy is again rather dependent on user experience and goals. For instance, an implementation of a de novo Galaxy platform more or less similar to our solution, is indeed possible, especially if using docker containers to run Galaxy under Windows or MacOS environments. However, even the easiest Galaxy implementation requires advanced bioinformatic skills to achieve the implementation; moreover, yet more important is having time enough for developing, installing, and maintaining the solution. Such a question is not trivial at all. For example, there is a Galaxy docker already assembled for analysis of RNA-seq data (https://github.com/bgruening/galaxy-rna-seq, accessed on 17 January 2023) and another for Variant-seq analysis (https://github.com/bgruening/docker-galaxy-exome-seq, accessed on 17 January 2023). However, these two dockers are not actively maintained or updated by the Galaxy community and, currently, both present significant absences in terms of tools and steps in comparison with the protocols we implement in our solution, which differ from Galaxy not only in the strategy of implementation, but also in how the implementation is made. Our solution is a Java implementation (shown in Figure 1) whose executables are available, either as desktop applications or as web applications. In contrast, Galaxy solutions are usually implemented as web applications, using Python. These differences of compilation extend not only to the client-side, but also to the server-side infrastructure, where the only common features are CLI tools dependencies. The rest of the contents (the collection of scripts to process the client queries, job, tracking system, API, etc.) are different implementations. The Galaxy implementation is Python, whereas ours is Java. It is commonly accepted that Java is faster and more efficient than Python because it is a compiled language while Python is an interpreted language. However, it is also widely accepted that Python is more concise in syntax, and therefore, easier to manage than Java. Hence, we cannot argue that one implementation is better than another but more properly that they are different implementations. Thus, some users will find advantages in our solution, while others will prefer Galaxy. Fourth, our solution and Chipster are certainly similar regarding context, operativity, and implementation, but they also differ from each other with respect to other aspects. Regarding scope, Chipster is currently available as an online web service allowing analyses for distinct -omics, altogether integrated in the same platform. This platform is hosted at the CSC’s server, a center of super-computation, and it is accessible at the following URL: https://chipster.csc.fi (accessed on 17 January 2023). However, Chipster is also available as a downloadable package that users can install and execute in their servers and PCs, albeit only under Linux OS environments, because the server-side bundle of Chipster only runs on this OS. We have no knowledge about the computational performance of the installable version of Chipster; however, being an open source package, users with programming skills can customize and optimize the platform and its performance if necessary. In contraposition, our solution is topic specific because in the GPRO suite each application is devoted to a particular topic (see also the introduction of this article). In this way, “*RNASeq*” and “*VariantSeq*” (the client applications of our solution) can be defined as the software implementation of two protocols of good practices: one for comparative transcriptomics (Figure 3) and the other for the characterization of SNPs and indels (Figure 4). This is a divide-and-conquer strategy with which we try to make our solutions not only more intuitive, but also highly efficient in their performance. Note that the more complex a bioinformatic platform is, with greater computational efforts and higher hardware resources, this platform will require the appropriate operation. In this sense, our solution has been extensively tested and validated and we can say that it runs satisfactorily in conventional PCs under any operative system (Windows, MacOS, or Linux), but it is also a powerful cloud resource that can be installed in remote servers. To this end, users can either use the desktop (RCP) version of “*RNASeq*” and “*VariantSeq*” executables that include a FTP browser to upload and download data from the PC to the server, or vice versa (see Figure 2). Alternatively, users can directly use the web (RAP) version of both applications that can be managed with the web browser.

In summary, our solution can be defined as a ready-to-use topic-specific platform for RNA-seq and variant-seq analysis that combines the user-friendliness, robustness, and security of desktop software, with the efficiency and versatility of cloud/web solutions to manage pipelines and workflows based on CLI tools. Our solution is accompanied by an AI system called GENIE that provides interactive support. To our knowledge, our solution is the first bioinformatic platform implementing an AI system device such as GENIE that integrates a jobs-tracking panel powered by an expert system and a chatbot (Figure 5) for users’ virtual assistance. GENIE is a newborn AI that logically needs more training to gain accuracy and efficiency. However, this is the reason why we have centralized the knowledge resources of GENIE online, as the idea is to allow this AI to learn from its interactions with all users. As for the GPRO suite, in this article we introduced “*RNASeq*” and “*VariantSeq*”, and the GSS platform that is portable via docker. In addition, we are preparing new publications for other applications of the suite (see the Introduction of this article for more details), but also new future implementations for “*RNASeq*”, “*VariantSeq*”, and the GSS. In the case of “*RNASeq*”, we aim to implement additional steps and new CLI tools, such as Seurat [63] and Crossmapper [64], to allow new pipelines and workflows to analyze single-cell RNA-seq and dual RNA-seq data. Regarding “*VariantSeq*”, we want to integrate new steps and tools from the GATK and other packages for the analysis of copy number variations (CNV), other calling functions from packages such as BCFtools [65], as well as new tailor-made tools for the filtering, prioritization, and annotation of variants. With respect to the GSS, we are preparing a new release of its docker with multiple user capabilities (the current docker version is limited to one or two users). As such, the GSS will require periodic updates to integrate the new releases, and its infrastructure will progressively increase in complexity and size. Thus, it is likely that we will eventually split the GSS into one docker per application to maintain the user-friendliness of this resource. Finally, it is also worth stressing that we are committed to implementing an interoperability layer that supports the Common Workflow Language (CWL) standards. This would allow a great level of extensibility to our software, in which users would be able to export the implemented workflows in “*RNASeq*” and “*VariantSeq*” to a portable format that they can import and use in a variety of platforms that support CWL. In this way, users will also be able to import their workflows from other platforms such as Galaxy to be used with “*RNASeq*” and “*VariantSeq*”, or in any other GPRO software-managing pipelines.

## Figures and Tables

**Figure 1 genes-14-00267-f001:**
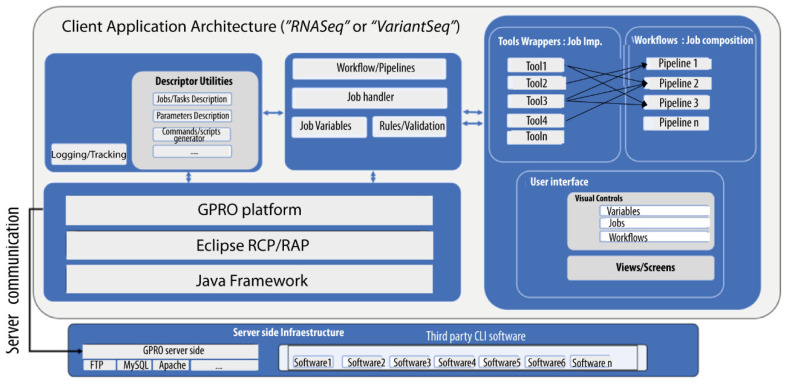
Client-side and server-side schematic and implementation. “*RNASeq*” and “*VariantSeq*” were both implemented using a common eclipse framework that enables encapsulation of third-party CLI tools as task wrappers, dynamically generated GUI views for each CLI tool, executable scripts, composable pipelines, and tracking/logging outputs of running jobs. The GSS provides the Linux environment and all other server requirements to run the CLI software (including scripts, R, Perl, Python, and MySQL server). Applications and the GSS connect via API.

**Figure 2 genes-14-00267-f002:**
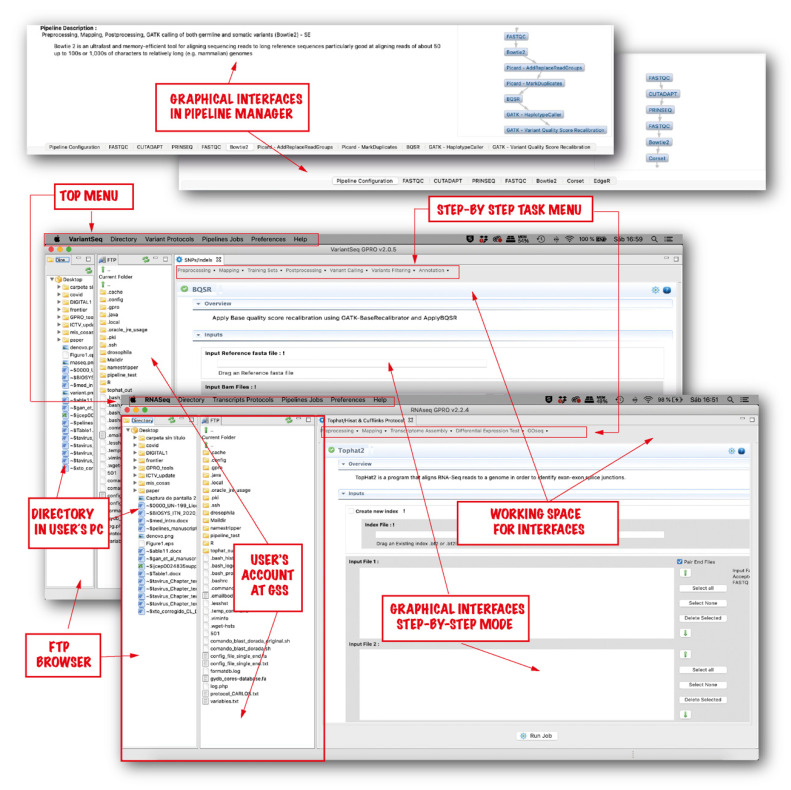
User interfaces of “*RNASeq*” and “*VariantSeq*”. Both applications have a common interface. The general procedure is as follows. Once an application has been linked to the GSS, the user should follow the subsequent steps. (first) Transfer the input files from the user’s PC directory to the GSS using the FTP browser. (second) Select the computational mode (it can be step-by-step or pipeline-like) (third) Drag the input files (fastq files, reference sequences, GTF/GFF files, Training Sets, etc.) from the GSS to the input fields of the selected interface/s. (fourth) Declare the output. (fifth) Set options and parameters. (sixth) Run the analysis.

**Figure 3 genes-14-00267-f003:**
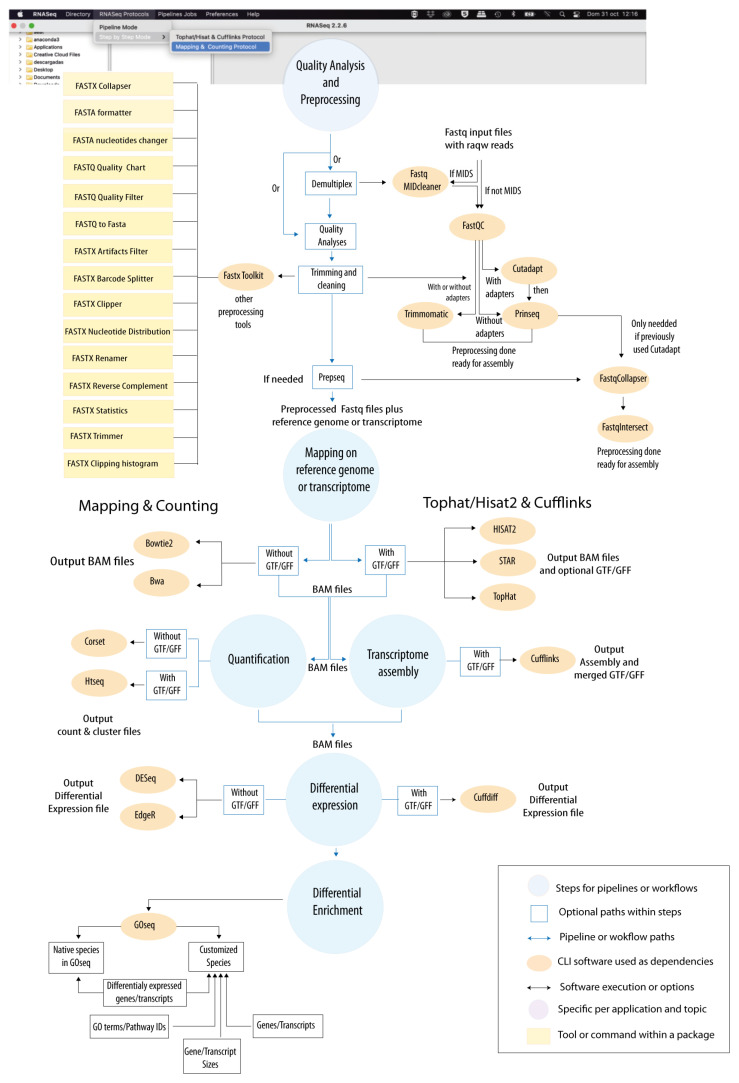
“*RNASeq*” protocol. Computational steps constituting the protocol of “*RNASeq*” for DE and enrichment analysis. The protocol is based on the following steps: “*Quality Analysis & Preprocessing*”; “*Mapping*”; “*Quantification*”; “*Transcriptome Assembly*”; “*Differential Expression*”; *and* “*Differential Enrichment*”. A summary of all CLI tools available for each step is provided in the figure. Two alternative paths (designated as “Mapping & Counting” and “Tophat/Hisat2 & Cufflinks”, respectively) are allowed. In Appendix A, we provide an example of GUI provided by “*RNASeq*” for the execution of CLI tools. A detailed tutorial for “*RNASeq*” usage is also available in the Data Availability section of this article.

**Figure 4 genes-14-00267-f004:**
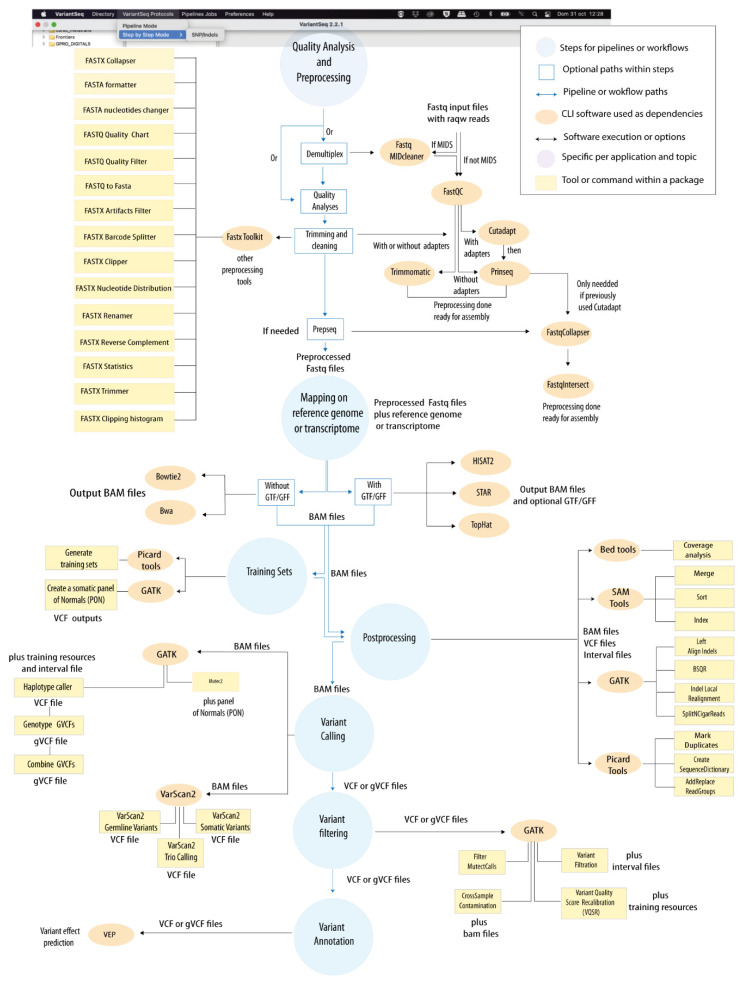
“*VariantSeq*” protocol. Computational steps within “*VariantSeq*” for calling and annotation of SNP and indels. This protocol is based on the following steps: “*Quality Analysis & Preprocessing*”; “*Mapping*”; “*Training Sets*”; “*Postprocessing*”; “*Variant Calling*”; “*Variant Filtering*”; and “*Annotation*”. Depending on the combination of tools selected in this protocol, users can call distinct types of variants (germinal, somatic, cancer, trio) from genome, exome, and transcriptome NGS data. In Appendix A, we provide an example of GUI implemented in “*VariantSeq*” for running CLI tools. In addition, we also provide a full tutorial for “*VariantSeq*” usage in the Data Availability section of this article.

**Figure 5 genes-14-00267-f005:**
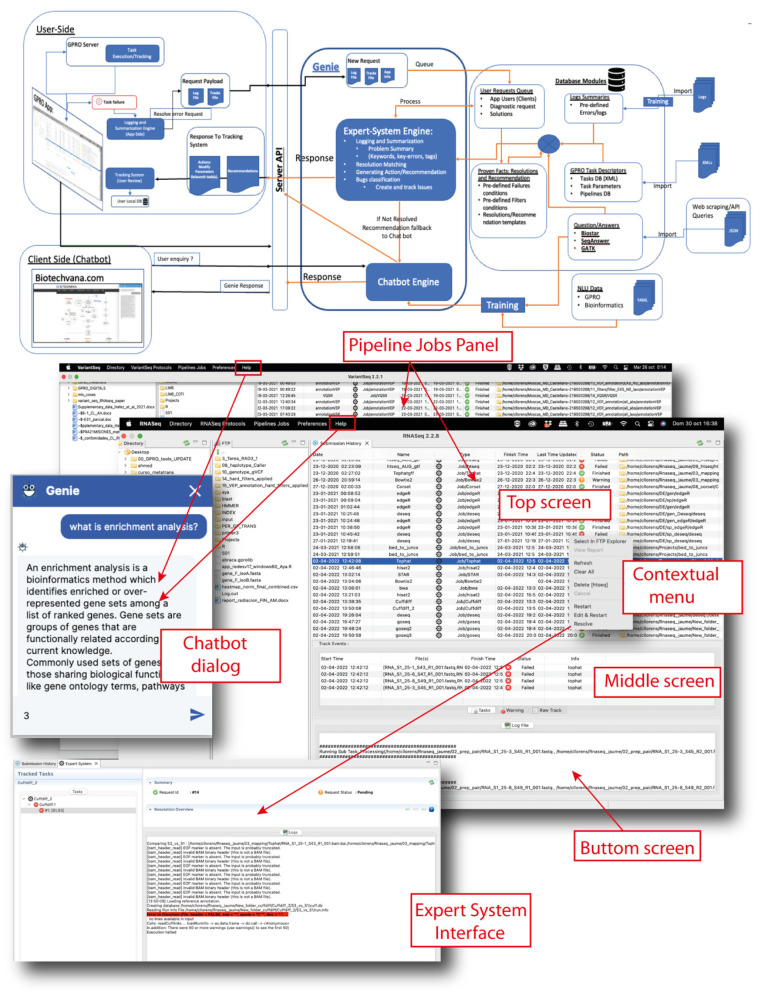
User support system. Schematic of the GENIE AI system. The chatbot and expert system engines and their knowledge databases are hosted on a remote GPRO server that communicates with the client applications and GSS via API. Each application presents a dialog available in the Help section of the Top Menu where the chatbot can be asked questions. Each application also has a pipeline jobs panel, which is a dynamic interface that summarizes all jobs summitted to the GSS and that provides information across three screens (Top, Middle, and Bottom) about the status of each specific job; specifically, if the job finished correctly (green icons), had some warnings (orange icons), or failed (red icons). By right clicking on the panel, a contextual menu will appear to provide tasks to manage the panel (described in text) and the expert system (shown at the bottom of the figure).

**Table 1 genes-14-00267-t001:** CLI software dependencies of “*RNASeq*” and “*VariantSeq*” at the GSS.

	CLI Third-Party Software	*RNASeq*	*VariantSeq*
Quality Analysis and Preprocessing	FastQC v0.11.5 [24]	**√**	**√**
	FastqMidCleaner 1.0.0	**√**	**√**
	Cutadapt 1.18 [25]	**√**	**√**
	PRINSEQ-lite 0.20.4 [26]	**√**	**√**
	Trimmomatic 0.36 [27]	**√**	**√**
	FastxToolkit 0.0.13 [28]	**√**	**√**
	FastqCollapser 1.0.0	**√**	**√**
	FastqIntersect 1.0.0	**√**	**√**
Mapping of Reference Genome or Transcriptome	TopHat v2.1.1 [29]	**√**	**√**
	Hisat2 2.2.1 [30]	**√**	**√**
	Bowtie2 2.2.9 [31]	**√**	**√**
	BWA 0.7.15-r1140 [32]	**√**	**√**
	STAR 2.7.0f [33]	X	**√**
Quantification	Corset 1.06 [34]	**√**	X
	Htseq 0.12.4 [35]	**√**	X
Post Processing	Bed Tools v2.29.2 [36]	X	**√**
	GATK v4.1.2.0 [37,38]	X	**√**
	Picard tools 2.19.0 [39]	X	**√**
	SAMtools 1.8 [40]	X	**√**
Transcriptome Assembly	Cufflinks v2.2.1 [41]	**√**	X
Differential Expression	DESeq 2.1.28 [42]	**√**	X
	EdgeR 3.30.3 [43]	**√**	X
	Cuffdiff v2.2.1 [41]	**√**	X
	CummeRbund 2.30.0 [44]	**√**	X
Enrichment Analysis	GOseq 1.40.0 [45]	**√**	X
Training Sets	GATK v4.1.2.0] [37,38,46]	X	**√**
Variant Calling	GATK) v4.1.2.0 [37,38,46]	X	**√**
	VarScan2 v2.4.3 [47]	X	**√**
Variant Filtering	GATK v4.1.2.0 [37,38]	X	**√**
Annotation of Variant Effects	Variant Effect Predictor 105.0 [48]	X	**√**

“**√**” means yes and “X” means not included. All the CLI software here summarized are integrated in the GSS docker image, except Varscan2 due to licensing questions. Authors interested in VarScan2 may find indications about how to install this tool in the GSS docker at https://gpro.biotechvana.com/tool/gpro-server/manual (accessed on 17 January 2023). Academic users can freely install it, while commercial users need to contact the VarScan2 authors to obtain the corresponding commercial license (for more details see https://github.com/dkoboldt/varscan/releases, accessed on 17 January 2023).

## Data Availability

RCP (Desktop) and RAP (cloud) executables of “*RNASeq*” and “*VariantSeq*” are available at: “*RNASeq*” Executable (https://gpro.biotechvana.com/download/RNAseq, accessed on 17 January 2023); “*VariantSeq*” Executable (https://gpro.biotechvana.com/download/VariantSeq, accessed on 17 January 2023). RCP versions are desktop executables for Windows, Linux, and/or macOS that can be installed on any PC, while RAP versions are cloud applications that can be managed with the web browser of any PC. Manuals of “*RNASeq*” and “*VariantSeq*” are available at: “*RNASeq*” Manual (https://gpro.biotechvana.com/tool/rnaseq/manual, accessed on 17 January 2023); “*VariantSeq*” Manual (https://gpro.biotechvana.com/tool/variantseq/manual, accessed on 17 January 2023). Tutorials for becoming familiar with the installation and usage of “*RNASeq*” and “*VariantSeq*” are accessible at: “*RNASeq*” Tutorial (https://gpro.biotechvana.com/software/RNASeq/RNAseq_tutorial.docx, accessed on 17 January 2023); “*VariantSeq*” *Tutorial* (https://gpro.biotechvana.com/software/VariantSeq/Variantseq_tutorial.docx, accessed on 17 January 2023). An image of the GPRO GSS Docker is available at the following URL: (https://hub.docker.com/r/biotechvana/gpro, accessed on 17 January 2023). Its installation is easy but requires a third-party software installer called Docker Desktop that is available at the following URL: (https://www.docker.com/products/docker-desktop, accessed on 17 January 2023). To proceed with the installation of the GSS, open the Docker Desktop software and execute the following command on its terminal:
local_path="/path/to/local_home"GPRO_USER="myUserName"GPRO_USER_PASS="myUserNamePass"docker run -d -p 80:80 -p 20-22:20-22 -p 65500-65515:65500-65515 -v/path/to/local_home:/home/gpro_user biotechvana/gpro
Please note that the words “myUserName” and “myUserNamePass” above refer to the username and password that the user chooses to access the GSS. A web version of the chatbot of GENIE is available at the following URL (https://gpro.biotechvana.com/genie, accessed on 17 January 2023). Fastq files used in the above referred tutorials for RNA-seq and variant-seq analysis, were obtained from the SRA archive at the NCBI [53]. local_path="/path/to/local_home" GPRO_USER="myUserName" GPRO_USER_PASS="myUserNamePass" docker run -d -p 80:80 -p 20-22:20-22 -p 65500-65515:65500-65515 -v/path/to/local_home:/home/gpro_user biotechvana/gpro

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
