# Peer review of "Client Applications and Server-Side Docker for Management of RNASeq and/or VariantSeq Workflows and Pipelines of the GPRO Suite"

_genes, 2023, doi:10.3390/genes14020267_

Round 1
Reviewer 1 Report
Hafez and colleagues describe a software solution to perform comparative transcriptomics and analysis of variants.
The authors describe very thoroughly a software system that seems potentially useful. But that's all there is to it, a mere description of a software tool.
Firstly, there is very little context of where it fits in the vast universe of existing tools with the exception of briefly mentioning Galaxy. No mention of other tools such as Chipster, seqmonk, ugene, or any of the miriad of existing solutions, both locally or online. For example, there is already a docker image for a galaxy specialized in comparative RNA-Seq data analysis (even if not actively maintained any more). It should be made much clearer what is the niche (if any) that this new system will fill and advantages over existing solutions. For example, I don't see a major advantage over Galaxy. It is quite simple, using docker, to have a local instance of Galaxy with specialized tools and workflows for RNA-Seq or for Variant calling. At least, as simple as GPRO.
The current trend regarding workflow management is the support of standards such as the Common Workflow Language (CWL) to facilitate the reproducibility and shareability of workflows, in the spirit of FAIR (Findable, Accessible, Interoperable, Reusable). It should be a mandatory topic to be mentioned in this work.
A note: the sentence in lines 101-103 is an innappropriate self-citation, as it has no bearing to the current work. It should be removed in my view.
minor poitns:
line 72: "can only be achieve" should be "can only be achieved"
line 77: "Most those" should be "Most of those"
line 96: "GPRO a bioinformatic" should be "GPRO, a bioinformatic"
line 105: "The here introduced solution" although correct sounds convoluted. I suggest "the solution introduced here"
Author Response
Reviewer 1 comments and authors’ response:
Reviewer 1 general comment 1:
Hafez and colleagues describe a software solution to perform comparative transcriptomics and analysis of variants.
The authors describe very thoroughly a software system that seems potentially useful. But that's all there is to it, a mere description of a software tool.
Authors’ response
Reviewer 1 is right in the idea that the aim of the article is to introduce a bioinformatic solution for RNA-seq and Variant-seq analysis based on two software applications and a server platform. Said that, we certainly appreciate and thanks Reviewer 1 for his/her criticisms and suggestions that are indeed helpful and really for improving the quality and clarity of our article.
Reviewer 1 general comment 2:
Firstly, there is very little context of where it fits in the vast universe of existing tools with the exception of briefly mentioning Galaxy. No mention of other tools such as Chipster, seqmonk, ugene, or any of the miriad of existing solutions, both locally or online. For example, there is already a docker image for a galaxy specialized in comparative RNA-Seq data analysis (even if not actively maintained any more). It should be made much clearer what is the niche (if any) that this new system will fill and advantages over existing solutions. For example, I don't see a major advantage over Galaxy. It is quite simple, using docker, to have a local instance of Galaxy with specialized tools and workflows for RNA-Seq or for Variant calling. At least, as simple as GPRO.
Authors’ response
We have rewritten the introduction and also the discussion of the article according to the criticisms of this reviewer and also those of reviewer2. We have extended the edits to the discussion because part of what the reviewer is suggesting here is to discuss deeper our solution. Anyway, we have extensively edited both the introduction and the discussion and added more bibliography. Evidently, we cannot cite every tool available because this article is not a review article, but we cite and discuss other tools already available from the enterprise and academy, including Chipster and Unigene (as suggested by this reviewer) but not SeqMonk. This is just because although SeqMonk seem to be an excellent resource but a tool mainly focusing on visualization and downstream analysis of mapping data (in general) thus meaning that is a SeqMonk is not a resource comparable with our “RNASeq” or “VariantSeq” but one of the resources that our application would be able to integrate in their protocols as a series of extra steps for downstream analysis. Hope reviewer1 find now appropriate the new edits and contents.
Reviewer 1 general comment 3:
The current trend regarding workflow management is the support of standards such as the Common Workflow Language (CWL) to facilitate the reproducibility and shareability of workflows, in the spirit of FAIR (Findable, Accessible, Interoperable, Reusable). It should be a mandatory topic to be mentioned in this work.
Authors’ response
We agree reviewer1 in that it is important to mention the CWL in this article. We have rewritten the discussion of this article to mention the Common Workflow Language (CWL) in the article as this kind reviewer has proposed (lines 1046-1053). The applications “RNASeq” and “VariantSeq” belong to the GPRO suite and we are still introducing the distinct applications. Part of our current research for future implementations is to adapt our pipelines and workflows to the CWL. And this is for many reasons, some logically because the general reasons to facilitate the shareability of workflow but others because for us is also advantageous and productive to implement new workflows according to CW as it facilitates indeed the job of programmers. So we are indeed opened to adapt our pipelines to CWL in the incoming future.
Reviewer 1 general comment 4:
A note: the sentence in lines 101-103 is an inappropriate self-citation, as it has no bearing to the current work. It should be removed in my view.
Authors’ response
In our humble opinion this citation is needed because these two sentences constitute an explanation of what the second release of the GPRO project is. In the previous sentence, we introduce the first release and in this second sentence we explain that the second release consists of 6 applications (“SeqEditor”, “RNAseq”, “VariantSeq”, “DeNovoSeq”, “Worksheet” and “STATools”). As they are independent tools we are publishing them in separate papers because each is a complex tool specific of one particular topic. Two of them (”RNAseq”, “VariantSeq”) are introduced in the current paper because make sense to introduce them together because both process resequencing data, another (seqEditor) was already introduced in another paper and logically if we refer the application we should to cite it if it is published and the remaining three correspond with three tools of which we are preparing at least two other manuscripts for introducing them. As we said, in our opinion that is not inappropriate but more properly contextual because we need to properly introduce the reader with a few words in the GPRO project.
We thus think that the problem is that these two sentences where we refer Seqeditor are not straightforward. Reading these two sentences we have seen that they were some grammar thypos from our side that were derived from actions to save changes and corrections after copyediting. These thypos make the two sentences to lose their intended meaning that is merely to review in a few words the historical and context of the GPRO project. Just this. What we have done is to rewrite this part of the introduction in order to give it the intended meaning and besides have also extended it a little bit to also address a comment of reviewer 2 which wondered about why we selected the topics of RNA-seq or variant-seq for developing our tools and not others. Please read the edited text between lines 160 and 178 of the edited version (that provides the whole context now edited) and if after this, you still consider that is an inappropriate self-citation, we will remove the citation.
Following is our response to the minor point comments of reviewer1
Reviewer 1 Minor point 1:
line 72: "can only be achieve" should be "can only be achieved"
Authors’ response
Done in the manuscript (now is line 87)
Reviewer 1 Minor point 2:
line 77: "Most those" should be "Most of those"
Authors’ response
Done in the manuscript, the text has been fully edited so new there is a new meaning but the edit addressing this comment is in line 93.
Reviewer 1 Minor point 3:
line 96: "GPRO a bioinformatic" should be "GPRO, a bioinformatic"
Authors’ response
Done in the manuscript, the text has been fully edited so new there is a new meaning but the edit addressing this comment is in line 166.
Reviewer 1 Minor point 4:
line 105: "The here introduced solution" although correct sounds convoluted. I suggest "the solution introduced here"
Authors’ response
Done in the manuscript, the text has been fully edited so new there is a new meaning but the edit addressing this comment is in line 162.
Reviewer 2 Report
My detailed comments are in the attached file, please revise it, Actually authors have a nice paper, and can be improved.

Author Response
Reviewer 2 comments and authors’ response:
Authors’ general response to reviewer 2:
We appreciate and thanks the positive and constructive comments of this expert to our work that without a doubt are helpful for improving the quality of this article. We have rewritten the manuscript with particular focus in sections Introduction and discussion to improve the article according to the criticisms and/or suggestions of Reviewer 2 and reviewer 1. Following is a point by point response to the criticisms addressed by this reviewer in the attached file.
Reviewer2 comment 1:
Line 30 highlighted yellow -> which ones?
Authors’ response:
This question refers to the collection of CLI tools used as pipeline dependencies. It is not possible to cite every dependency in the abstract because they are 30 CLI tools but they are summarized and properly cited in table 1 and also mentioned in figures 1 and 2.
Reviewer2 comment 2:
Line 41 highlighted yellow -> This is amazing!
Authors’ response:
Thank you very much for the positive comment and appreciation, we also think is a very innovative element that could be helpful for the users precisely because NGS data analyses involve a lot of concepts, protocols, tools, and formats to manage.
Reviewer2 comment 3:
Line 48 highlighted yellow -> what is the purpose of this keyword??
Authors’ response:
Resequencing word is used as a key word because the NGS data traditionally used to perform RNA-seq and Variant-seq analyses derives from sequencing experiments based on genome/s has been with available reference sequence. This is more explicitly detailed later in the introduction of the article.
Reviewer2 comment 4:
Line 61 highlighted yellow -> can you add a reference for this?
Authors’ response:
Done in text. Please note that after edits this phrase and the citations are now in line 77 .
Reviewer2 comment 5:
Line 74 highlighted yellow -> absolutely correct!
Authors’ response:
We agree in the same idea with this kind reviewer. In fact, this perception is what motivated us to develop tools (in the context of the GPRO project) characterized by the simplicity and the friendliness of use and as unfortunately NGS is not easy at all) provide also additional tools of support like the chatbot.
Reviewer2 comment 6:
Line 148 highlighted yellow -> How about processors? this software works in a quite old mac with intel processor
Authors’ response
It should work with at least mac intel processors since 2011 but It is difficult to say because that depend on a combination of factors. First there is some minimum software requirements, for the client applications, Java 11 is required, so you need OS with Java 11 support. As for running docker there is also some software requirements that the OS must meet. If the old Mac can run an OS with those software requirements. It can run the clients’ apps and the Docker with no problem. The client desktop application has a small footprint in term of memory use and processing requirements. the GSS by itself also does not require too much RAM, 2G of RAM can be sufficient for running the services. However, what requires more Memory and processing costs is the bioinformatic third party tools and pipelines launched be the GSS. And the requirements vary from one tool and another and from dataset to another. So working with human genome normally require a lot of RAM while working with yeast genome not. So based on our experience we recommend 500G storage and 16Gb of RAM but this could be more or less the actual requirements for your dataset. Anyway, if you are interested in installing it in a very old mac, please try it and if you have some troubles please contact us to see if we help.
Reviewer2 comment 7:
Table 1 footer highlighted yellow -> is there any substitute for Varscan 2 that does not have licensing questions?
Authors’ response
GATK and Varscan are perhaps (and this is always said since our current opinion and humble experience about the topic) two callers most widely used in pipeline for variant-seq analysis. This is the reason we implemented both. GATK is a perfect substitute of varscan and the license issue is not a problem at all because its installation is very easy and it is well explained in the web site of the GPRO Server side plataform (https://gpro.biotechvana.com/tool/gpro-server/manual). It is just that according to this license academic authors can install and use this tool without restriction and industrial users need to ask the original authors for a license of use. We thus think that the easiest way to implement Varscan is to provide an easy step for its installation and leave the users to decide what to do if installing it or not. Of course, there more and excellent packages to perform variant analysis. We plan to implement other calling tools from packages like bcftools and others like perhaps freebayes in further releases of variant-seq. In fact, this first version is limited to SNPs and Indels, but we also aim to add callers of viral variants, CNVs and other mutations. We will try to make these implementations in the next/s release/s of variantseq.
Reviewer2 comment 8:
Line 172 highlighted yellow -> what about stackoverflow?
Authors’ response
Excellent suggestion. We (at least me, the corresponding author) didn’t know stackoverflow. We mainly focused on pubmed because it is specialized biological/biomedical literature and therefore also cover bioinformatics and the social networks like GATK forum, SeqAnswers and Biostar because they are the most comprehensive forums of experts for addressing questions and answers to resolve bioinformatic issues. However, we will check it out stackoverflow to see if we can also obtain some information to improve the knowledge databases that feed the chatbot and the expert system (because they are continuously updated) or for redirecting the users for any specific problem to that forum if it provides a solution.
Reviewer2 comment 9:
Lines 199-200 highlighted yellow -> did the authors have a plan to improve their chatbot with another languages like chinese or spanish??
Authors’ response
We have prepared this first release of the chatbot in English because obviously, science mainly speaks English. However, and effectively, we are planning to extend the chatbot to other languages like at least Spanish, Chinese and Arabian where we will probably invite other researchers of other countries to collaborate with us for improving the power and language extent of this AI.
Reviewer2 comment 10:
Line 217 highlighted yellow -> comment in blank
Authors’ response
This phrase has been highlighted but there are no comments associated to this phrase. A priori we do not find any problem in this phrase but we have edited anyway it a little bit to be more specific, by changing
the original phrase
“RNASeq” and “VariantSeq” are two cross-platform client applications built for the processing and analysis of resequencing data obtained via NGS technologies.”
substituted by the following one
“RNASeq” and “VariantSeq” are two cross-platform client applications for analysis of resequencing data obtained via NGS technologies.
If there is any other suggestion here, we will be please to follow it. Please note that after edits this phrase and the citations are now in lines 577 and 578
Reviewer2 comment 11:
Line 272 highlighted yellow -> The interface from this image is from OS system, the software proposed in this manuscripts also has an interface for Windows or Ubuntu?
Authors’ response
Yes, there are windows, Macos, and Linux executables for both applications. In the Section Data availability, we provide the links to access them. In addition, there are also RAP (web) versions of each application too that can also be downloaded at the same URLs.
Reviewer2 comment 12:
Line 280 highlighted yellow -> Did the authors revise the webserver "ideamex" to check the good practices in RNASeq.
http://www.uusmb.unam.mx/ideamex/
Authors’ response
Yes, we did, ideamex follows a pipeline protocol based only on the differential expression step (not the other steps) based on tools for differential expression like Deseq and EdgeR. We have this step implemented in the protocol of “RNASeq” the mapping and counting path. Additional to this, the protocol of “RNAseq” also gives the option to make the differential expression using Cufflinks by accessing the Tophat/cufflinks path of the protocol. In addition, we also provide a last step for GO enrichment.
Reviewer2 comment 13:
Line 304 highlighted yellow -> did the authors provide any example to run this protocol "RNASeq"??
Authors’ response
Yes, Supplementary file S2 provides an example of GUIs implemented in “RNAseq”. Also, Supplementary file S4 is a dynamic GIFs illustrating how to manage the pipeline mode of “RNASeq”. Finally, the Section Data Availability provides a tutorial with detailed instructions to run the “RNASeq” protocol using the two step-by-step and pipeline modes.
Reviewer2 comment 14:
Line 327 highlighted yellow -> the same as my previous comment, did the authors provide any example to run this protocol?
Authors’ response
Yes, same answer to that for “RNASeq” comment 13. In the case of VariantSeq, Supplementary file S3 provides an example of the typical GUI of VariantSeq in the step by-step mode. Also, Supplementary file S5 is dynamic GIFs illustrating how to manage the pipeline mode of “VariantSeq”. Finally, a tutorial with details about how to run the “VariantSeq” protocol using the two step-by-step and pipeline modes, is also provided in Section Data Availability.
Reviewer2 comment 15:
Line 340-341 highlighted yellow -> Great! also the authors can add this sentence in the figure captions of figures 3 and 4
Authors’ response
Done in manuscript. The captions of figures 3 and 4 include a sentence where Supplementary files S3 and S4 and respectively referred. In addition, each caption also informs of the availability of a full tutorial in the Section Data Availability.
Reviewer2 comment 16:
Line 348 highlighted yellow -> After?
Authors’ response
Done in manuscript. Please note that after edits this phrase and the citations are now in line 772.
Reviewer2 comment 17:
Line 356 highlighted yellow -> what was the motivation to select these datasets?.. any reason in special?
Authors’ response
In our humble opinion a tutorial should be created using trusted or well-known material. That’s all. We selected these datasets because they were previously analysed by us in previous studies and in that way we were able to test and validate each tool implemented in “RNASeq” and “VariantSeq”. Part of the multiple distinct results used to validate each tool have been collected together to create a tutorial of usage for each application. Fastq libraries proposed by each tutorial are all publicly available in the SRA archive which guarantee the reproducibility. In the case of genomes, same question. For The “VariantSeq” we use the human genome grhg38 release. For the RNASeq tutorial we use a genome release of Sparus aurata publicly available in the CSIC (the research council of spain) but we also refer the possibility to use another release of the S. aurata genome available at the NCBI.
Reviewer 2 comment 18
Line 397 highlighted yellow -> what were the main advantages to select GENIE as AI systems, for this project?
Authors’ response
The AI GENIE module was not selected but that is has been created de novo using phyton and two packages like Django and Rasa. The advantage of using Phyton is that a very accessible programming language with multiple tool options so it is ideal for this kind of approaches. Django is a framework from deployment web applications and RESTful API as it is written in python, it enabled us for full integration of the machine learning methods used to build GENIE AI and the chat modules. Rasa is the framework work from building and training chatbot as it is also developed with python so it fits perfectly in the whole AI module. The response is thus that Python, Rasa and Django complement each other satisfactorily in order to try AI plus machine learning approaches like this one.
Reviewer 2 comment 19
Line 414 highlighted yellow -> why the authors select these analysis instead or something related to proteomics or another "omic" science?
Authors’ response
We have rewritten the introduction of this article to explain with more detail about the GPRO project which it could be important to clarify why these topics and no others. Basically, “RNASeq” and “VariantSeq” have been developed in the context of the GPRO suite project (https://gpro.biotechvana.com). Each application we develop in the context of that project devoted of a particular topic in the fields of genomics and transcriptomics. The current suite is composed of 6 applications being each one a complex tool that is independent of the others. For this reason, we are publishing them in separate papers. In a former paper, we published one of them seqeditor (that is a tools for sequence analysis and genome browsing). In this article, we are publishing “RNASeq” and “VariantSeq” which are pipeline managers for differential expression and variant calling analysis based on resequencing data. The next article we are preparing is for introducing another application called DeNovoSeq that is devoted for the assembly and annotation of genomes and transcriptomes characterized for the first time (i.e. de novo characterization). The two other applications of the suite are named Worksheet and STATools because they are devoted for integrative and statistic and metagenomic analysis, respectively. With these six tools we cover a significant extent in genomics but of course not all. Besides of the updated we are planning for the current applications of the suite we have also in mind to develop at least three new applications within the suite: one will focus on Methylseq analysis whereas we think that from that point on, we won’t need to develop more applications oriented to Genomics (just periodically update the current ones according to the evolution of the state of the art in genomics). For this reason, we are also currently investigating how to address other omics topics like flow cytometry and proteomics within the GPRO project and that correspond with the two applications we have in our pipeline of things-to-do in the incoming future. In the case of flow cytometry, we are starting. We are in a phase of exploration and planification of the protocol, collaborating with some students and other colleagues to put some very early ideas in practice like prototypes (see for example https://github.com/aligogon/Cytometry-Biotechvana). Same strategy is scheduled for proteomics.
Reviewer 2 comment 20
Line 477 highlighted yellow -> this is amazing and i think, this is a notable insight of this work
Authors’ response
Thank you very much for your positive feedback. The idea of implementing the chatbot in other languages is certainly exciting. Thank you for this suggestion.
Reviewer 2 comment 21
Line 482 highlighted yellow -> which ones?, please clarify
Authors’ response
Done in the manuscript. We have added four new citations regarding further implementations we are preparing for “RNASeq” and “VariantSeq”. Please note that after edits this phrase and the citations are now in lines 1035-1040.